# Study on the Limit of Moisture Content of Smoldering Humus during Sub-Surface Fires in the Boreal Forests of China

Sainan Yin [1], Yanlong Shan [1,*], Shuyuan Tang [1], Grahame Douglas [2], Bo Yu [1], Chenxi Cui [1] and Lili Cao [1]

[1] Science and Technology Innovation Center of Wildland Fire Prevention and Control, Beihua University, Jilin 132013, China

[2] School of Engineering, Design and Built Environment, Western Sydney University, Coffs Harbour, NSW 2450, Australia

* Correspondence: shanyl@ustc.edu.cn; Tel.: +86-186-0449-8158

**Abstract:** A sub-surface forest fire is a kind of fire that spreads slowly with no flames and lower temperatures, and threatens the ecosystem and human life. The moisture content of humus is considered to be an important factor in determining fire occurrence and sustaining. The humus of the *Larix gmelinii* in the Daxing'an Mountains was selected for the experiment, the limit moisture content condition of sub-surface forest fires was determined by an experiment simulating smoldering, and the prediction model of the probability of sub-surface forest fire occurrence was established. The results will be of great significance for the prevention, monitoring, and fighting of sub-surface forest fires in the boreal forest. The results showed that when the moisture content of humus in the upper layer was low, the smoldering process could be self-sustaining at 20%. For deeper layers of a depth of 18 cm, this increased to 30% moisture content of the humus and was the critical depth for sub-surface fires. The moisture content of 40% was a limit to burning where smoldering can only last for a short duration and is then extinguished. When the moisture content of the humus was 20%, the smoldering temperature was higher and the rate of spread was faster, with smoldering being maintained for longer periods at 30% moisture content. The regression prediction model of the highest temperature and vertical rate of spread in a column of humus was correlated to moisture content and depth, and the model significance was good at $p < 0.01$. Based on moisture content and depth, the occurrence probability prediction model of sub-surface fires has a good correlation ($R^2 = 0.93$) and high prediction accuracy (AUC = 0.995). The effect of moisture content (Or = 4.008) on the occurrence probability of sub-surface fires is higher than that of depth (Or = 2.948). The results point out that it is necessary to prevent and monitor the occurrence of sub-surface fires when the humus moisture content is less than 40%. In order to reduce the risk of sub-surface fires, the monitoring time of the fire field should be extended after the fire is extinguished due to the slow-burning process of the sub-surface fire. Increasing the moisture content of the humus is an important method to reduce the probability and restrain the spread of sub-surface fires.

**Keywords:** smoldering of sub-surface fire; moisture content limit; humus; occurrence probability prediction; Daxing'an Mountains





## 1. Introduction

Sub-surface forest fires mainly occur in the humus layer (duff) and peat layer, which spread slowly with no flames and lower temperatures. This leads to greenhouse gas emissions, a global ecosystem problem [1,2]. The northern temperate, subtropical, and tropical climatic regions are the areas of note for sub-surface fires [3]. Recent research shows that there are also large sub-surface fires in the Arctic [4]. Drought is a pre-conditioning factor and is caused by natural and human factors, and increases the risk of sub-surface fires [1,5]. In recent years, the release of greenhouse gases by human activities has kept on rising, and climate change is accelerating. The ambient temperature in northern areas has

increased by 0.44 °C per decade [6,7]. Global warming, therefore, creates conditions for the increased occurrence of sub-surface forest fires, which also increases the occurrence, frequency, and associated harm associated with such fires [8–10]. Sub-surface forest fires adversely affect the physical and chemical properties of soil and hydrological conditions, destroy plant roots, and cause large areas of tree collapse and tree death. These sub-surface fires in the northern forests of China will also thaw the frozen soil and affect the overall forest community [11–13]. Sub-surface forest fires release large amounts of greenhouse gases and associated toxic gases [14,15]. The large-scale peat fires in Indonesia, which occurred in 2019, resulted in serious smoke haze in Indonesia and surrounding countries [4]. As such, it can be inferred that these sub-surface fires are a serious risk to the local ecosystem as well as human life and property.

These smoldering sub-surface fires are extremely complex processes that can last for weeks, months, or even longer once initiated [16]. Researchers have found that oxygen content, moisture content, soil structure, and inorganic content are the main contributing factors in the occurrence and development of sub-surface fires [3,17–20]. The moisture content of combustible organic matter is considered a determining factor in the occurrence and maintenance of combustion [21,22]. At present, it is the main method for large-scale forest fire occurrence prediction and fire behavior research to calculate the moisture content of fuel by the remote sensing estimation method and meteorological element regression method [23–25]. These threshold values for sub-surface forest fires are an area of new research [4]. Studies show that the maximum depth of continuous smoldering can reach more than 5 m [3], and with global warming, the carbon loss caused by sub-surface fires in the boreal peatland will reach up to 5.44 million tons by 2100 [4]. Sub-surface fires are easier to ignite and sustain with a peat moisture content of less than 60% [26]. Although the threshold value of sub-surface forest fires has attracted attention in recent years, there are few quantified studies, especially in the humus fires of the boreal forest.

Humus or peat is composed of decomposing plant material under wet and acidic conditions, and is rich in carbon [27]. Both existing surface fires and lightning strikes can initiate sub-surface fires. Due to the influence of climate and geography, the Daxing'an Mountains are the area with the greatest incidence of sub-surface forest fires in the coniferous boreal forest of China, and the frequency of these sub-surface fires in this region has been increasing in recent years [28]. However, there is little research on these sub-surface fires in this region. This paper aimed to quantify the limit moisture content condition of the sub-surface forest fires according to the simulating smoldering experiment of *Larix gmelinii*, which is the typical fuel in the Daxing'an Mountains, and revealed sub-surface fire characteristics under different moisture contents and established the occurrence probability prediction model of sub-surface forest fires.

## 2. Methods

### 2.1. Study Area

The study area is located in the Bila River National Nature Reserve of the Daxing'an Mountains in Inner Mongolia (123°04′28.9″~123°29′16.1″ E, 49°19′39.5″~49°38′29.7″ N). This administrative region belongs to the town of Nuomin, Hulunbuir City, in the Inner Mongolian Autonomous Region. This region has a middle temperate zone within a humid and sub-humid continental monsoon climate. Temperature varies greatly; spring is dry and windy, summer is hot and rainy, temperature obviously changes in autumn, and winter is cold and dry with an annual average temperature of −1.1 °C. The annual precipitation is 490–560 mm, the annual accumulated temperature is 2014.4 °C, and the frost-free period is 90–115 days. The vegetation is mainly coniferous and broadleaved mixed forest. *Larix gmelinii* is the dominant species, and is associated with *Betula platyphylla*, *Betula dahurica*, and *Populus davidiana*. The reserve is dominated by virgin forests and wetlands; forest fires have been frequent in the past, especially the super-hot forest fires that occurred in 2017. The thicker humus layer and higher carbon content in the reserve provide the conditions for the occurrence of sub-surface forest fires.

## 2.2. Sampling and Processing

This study of the *Larix gmelinii*, a typical flammable species in the Bila River National Nature Reserve, uses five sample plots of 30 m × 20 m under the forest canopy with three quadrats of 50 cm × 50 cm on the diagonal of each sample plot (Figure 1). Samples of the humus layers in these quadrats were then analyzed in the laboratory. The upper layer of combustible materials is easily affected by climate and the local environment, resulting in low moisture content and high flammability during forest fires. The lower layer combustibles of the samples are higher in moisture content. When the upper layers are ignited, smoldering can progressively spread downward at a slow rate [29].

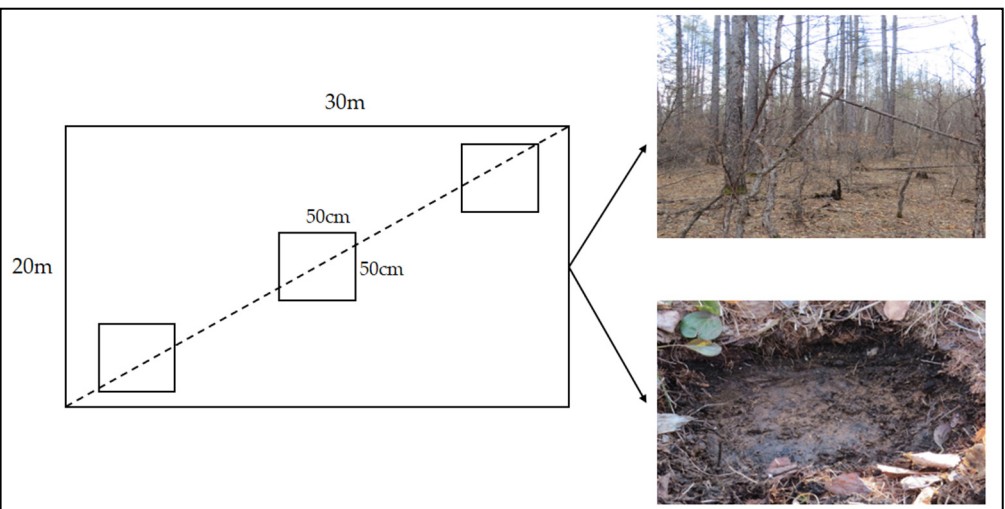

**Figure 1.** Humus collection.

In order to simulate actual sub-surface fires, the sample moisture content of the upper layer was set at 15%, with the moisture content gradient of the lower layer being set at 20%, 30%, and 40%, respectively. The humus was placed in a cool and ventilated place to dry naturally. The moisture content was measured with a halogen lamp rapid moisture monitor (gravimetric method for the moisture content, (wet weight − dry weight)/wet weight)) every 6 h, until the appropriate moisture content was obtained. Samples of combustibles with different moisture contents were placed in sealed plastic bags for experimental conditions. Humus moisture content was measured with three samples prior to the smoldering experiment using the rapid moisture monitor, with the average value being used as the experimental moisture content of the humus samples.

## 2.3. Simulating Smoldering Experiment

The smoldering experiment uses a single dimensional vertically experimental cylindrical container (Figure 2), which is suitable for the study of rate of downward spread during the smoldering period [19,30]. This cylindrical smoldering container has dimensions of 30 cm in height, 10 cm in bottom thickness, 10 cm in wall thickness, and 10 cm in internal diameter. The material of the container is an aluminum silicate ceramic fiber, which has good heat-retaining properties. Type-K thermocouples with a length of 30 cm and a diameter of 2 mm were used to measure temperature changes in the smoldering humus, and the collected data were transmitted to a laptop computer through a data acquisition module composed of a 16-channel NI9213 voltage acquisition board and DAQ-9174 chassis (4 slots) produced by American NI Company (detection precision of temperature <0.25 °C). The data acquisition software was Labview2018, which can record the temperature change curves collected by the thermocouples. An infrared heating plate (30 cm in length, 20 cm in width, 5 cm in height) was used as the ignition source, and a temperature control meter was connected between the heating plate and the power supply to keep the temperature of

the heating plate at a constant temperature of 500 °C, which was preheated for 2 h, prior to placement on the samples.

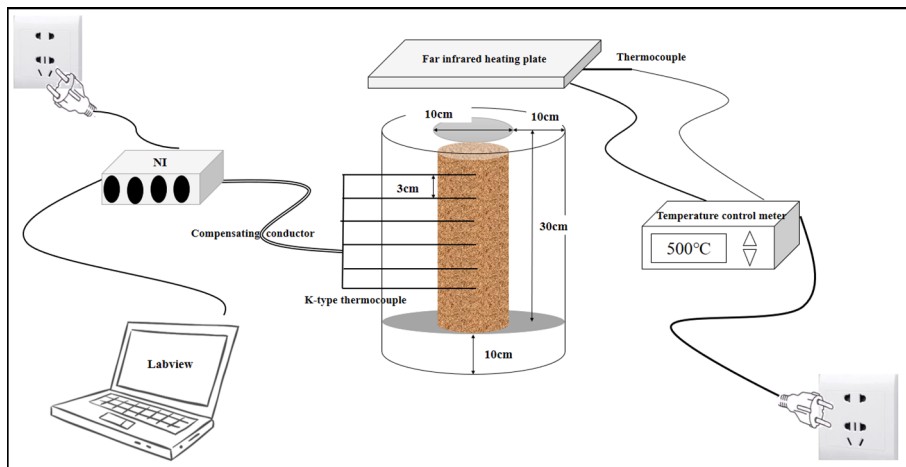

**Figure 2.** The design of the simulated smoldering experimental equipment.

Humuses with different moisture contents were then placed into the smoldering furnace, respectively. A small hole was drilled every 3 cm on the side of the furnace, and a thermocouple was inserted into the middle of humus. The thermocouples and data acquisition module were connected by compensation wires, and the data were transmitted back to the computer with a frequency of 10 s in duration. The temperature curve of each thermocouple was generated on the computer, and the data were stored. The experiment was repeated five times for each moisture content gradient.

Figure 3 shows the process (stages) of smoldering during the experimental simulation. Humus consumed in the upper layers through combustion was a mixture of ash and carbon black (See Figure 3D below).

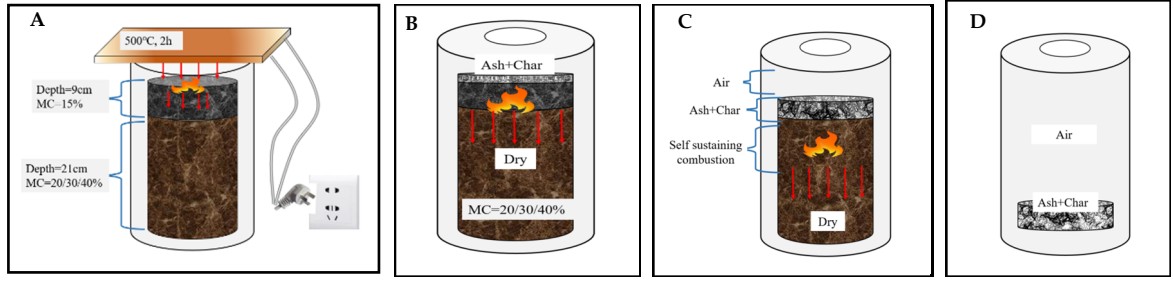

**Figure 3.** Process of the smoldering samples ((**A**): Igniting; (**B**): Start self-sustaining combustion; (**C**): Self-sustaining combustion; (**D**): Completely extinguished).

### 2.4. Data Processing and Analysis

Two-way ANOVA was used to analyze the influence of moisture content and depth on the peak temperature, time (the period needed to reach the peak temperature), and rate of spread of the smoldering samples (significance level $p < 0.05$). A regression analysis was used to establish the relationship between the moisture content, depth and peak temperature, and rate of spread. A draw box diagram was used to present the results for comparison, and represents 20%–80% of the experimental data. The inner line of the box is the mean value. The upper and lower extension lines are the maximum and minimum values. The difference is not significant if there is any same lowercase letter within the Box figures. The occurrence probability prediction model was based on a logistic regression model [31,32].

The dependent variable of the logistic regression model is discontinuous, which could be a binomial or multinomial function. Independent variables could be continuous

variables or categorical variables [33]. If the occurrence probability of sub-surface fire is *P*, then the nonoccurrence probability of sub-surface fire is $(1 - P)$ and is expressed as [34]

$$ln(\frac{P}{1-P}) = \beta_0 + \beta_1 x_1 + \beta_2 x_2 + \ldots + \beta_n x_n \tag{1}$$

where $\beta_0$ is constant, independent variable, $x_n$ is driving factors, $\beta_n$ is the coefficient of their variables.

The ROC (receiver operating characteristic) has been widely used in the evaluation of forest fire occurrence probability prediction models. AUC (area under the curve), the area under the ROC curve, can be used to evaluate the accuracy of the prediction model. The value of AUC ranges from 0.5 to 1. The higher the value of the AUC, the better the sensitivity, specificity, and accuracy of the prediction model [35].

## 3. Results

### 3.1. Limit Moisture Content of Sub-Surface Fire Smoldering

During the smoldering simulation experiment, when the humus in the upper layer was relatively dry, the sub-surface fire was self-sustaining during the whole smoldering process, and the combustion could last 25–35 h where the humus moisture content in the deep layer was 20% or 30%. The highest temperature recorded with 20% moisture content was 658.35 °C (27 cm), with the temperature staying above 300 °C for 19.13 h (27 cm). The highest temperature recorded with 30% moisture content was 639.23 °C (27 cm), with a temperature above 300 °C for 22.47 h (27 cm) (see Figure 4A,B).

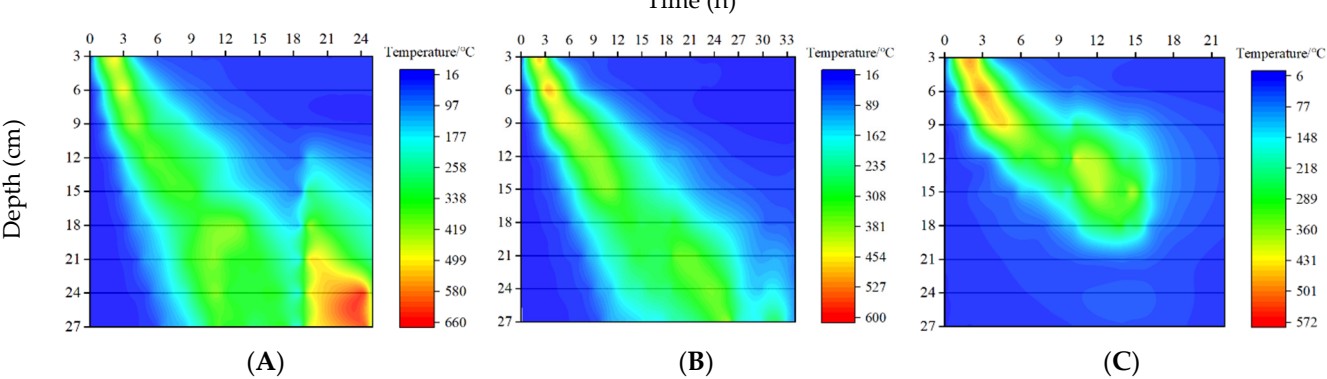

**Figure 4.** Smoldering characteristics of humus with different moisture content (samples (**A**–**C**): Smoldering temperatures of lower humus depths with moisture content of 20%, 30%, and 40%).

According to Section 2.3 (Figure 3C), after the humus with the lower moisture content in the upper layer was ignited, the moisture content would progressively dry out before burning into the lower position with the higher moisture content. The duration for the combustion to reach the highest temperature was long (Figure 4A,B). The temperature in the upper layers dropped quickly with combustion, while the temperature in the deeper layers was high and lasted for a long time until consumed.

When the humus moisture content of the deep layer was 40%, the smoldering could only be maintained for a short period and spread to a depth of 18 cm. The highest temperature for this sample was 492.41 °C (12 cm), indicating that 40% was the limit of moisture content for the smoldering experiment (Figure 4C).

The spread rate of smoldering in the upper humus with lower moisture content fluctuated greatly, while the spread rate in the deep humus with higher moisture content fluctuated less. Smoldering in the humus with a moisture content of 20% spread fastest at a depth of 12 cm (1.77 cm/h); then the spread rate slowed down, and the spread rate at a depth of 18 cm was only 1.52 cm/h. Smoldering in the humus with a moisture content of 30% spread fastest at a depth of 15 cm (1.33 cm/h); then the spread rate and the fluctuation

slowed down with the increase in depth, the lowest spread rate at a depth of 21 cm being only 1.21 cm/h. The fastest spread rate was detected at a depth of 18 cm (1.60 cm/h) with a moisture content of 40% (Figure 5).

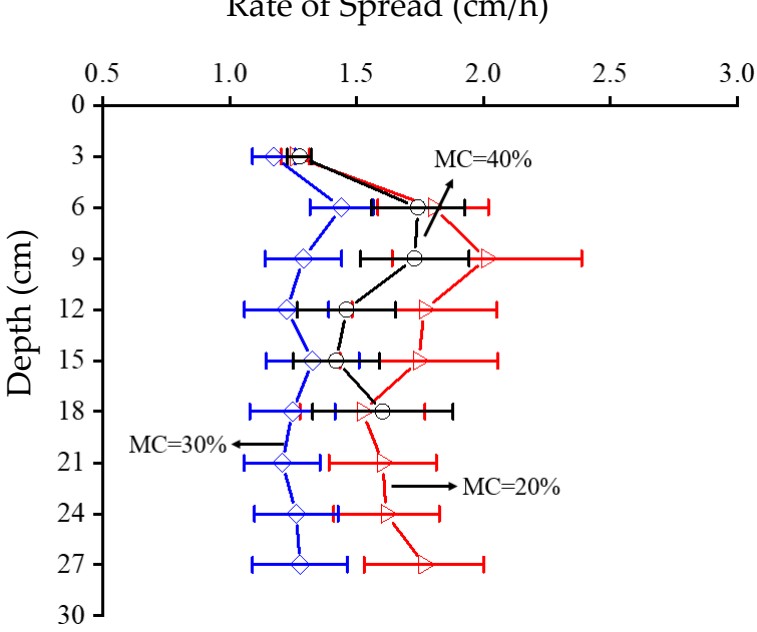

**Figure 5.** Average (and range) rates of spread of samples for depth of beds (cm).

### 3.2. Factors of Smoldering

Humus moisture content and depth both had a significant influence on the highest temperatures and period of smoldering ($p < 0.01$), but there was no significant difference in the influence of interaction between them ($p > 0.05$). The rate of spread for smoldering had a significant difference only among the different moisture contents ($p < 0.01$) (Table 1).

**Table 1.** Variance test on the influence of humus moisture content and depth on smoldering characteristics during sub-surface fires.

|  | Factor | *df* | *F* | Significance |
|---|---|---|---|---|
| | Moisture content | 2 | 22.87 | <0.01 |
| Peak temperature | Depth | 5 | 10.59 | <0.01 |
| | Moisture content × Depth | 10 | 2.01 | 0.052 |
| | Moisture content | 2 | 5.89 | <0.01 |
| Time | Depth | 5 | 8.46 | <0.01 |
| | Moisture content × Depth | 10 | 0.35 | 0.91 |
| | Moisture content | 2 | 5.75 | <0.01 |
| Spread rate | Depth | 5 | 0.14 | 0.97 |
| | Moisture content × Depth | 10 | 0.13 | 0.99 |

The highest temperature was exhibited for 20% moisture content, and there was a significant difference with the other moisture contents with increased moisture content. The period to reach the highest temperature was for the 30% moisture content, and there was a significant difference with the other moisture content samples (Figure 6A,B).

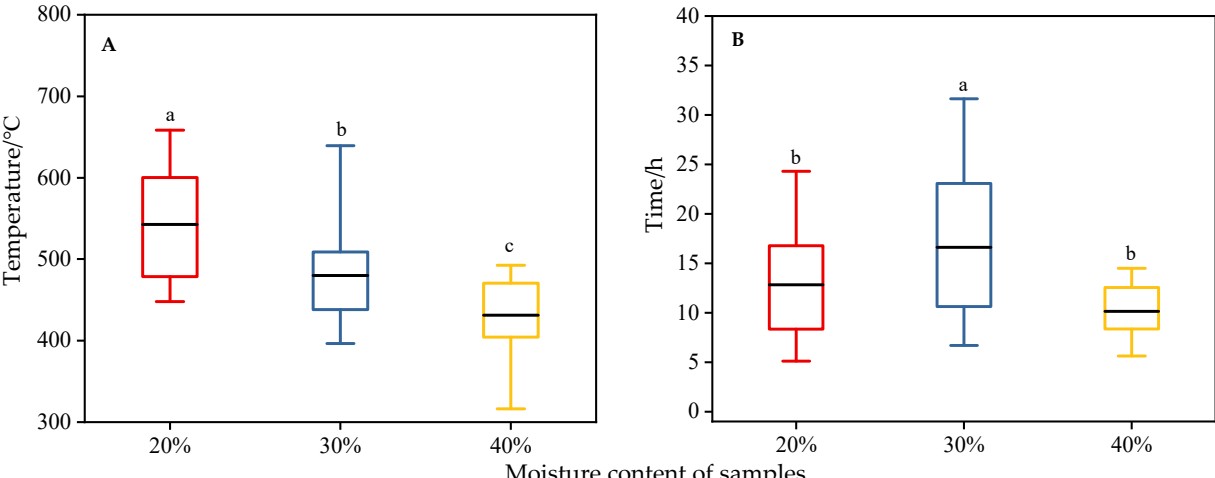

**Figure 6.** Variance analysis on influence of moisture content on temperature and combustion period during sub-surface fires ((**A**): Effect of moisture content on peak temperature; (**B**): Effect of moisture content on time; a–c means that if there is any same lowercase letter, the difference is not significant).

The highest temperature varied greatly at different depths, with the average highest temperature at a depth of 18 cm, which also was the lowest depth recorded (Figure 7A). The humus in deeper layers took the longest time to reach the peak temperature (27 cm), and there was little difference between the combustion times at the depths of 12–18 cm. The time to the peak temperature at a depth of 18 cm and 21 cm was significantly different; the longest smoldering time from 18 cm to 21 cm was 2.51 h (Figure 7B). When the humus in the upper layer was completely burned, the temperature of the humus with higher moisture contents in the deep layer initially decreased for a period and then showed a rising trend, indicating that the smoldering fire had begun to self-sustain.

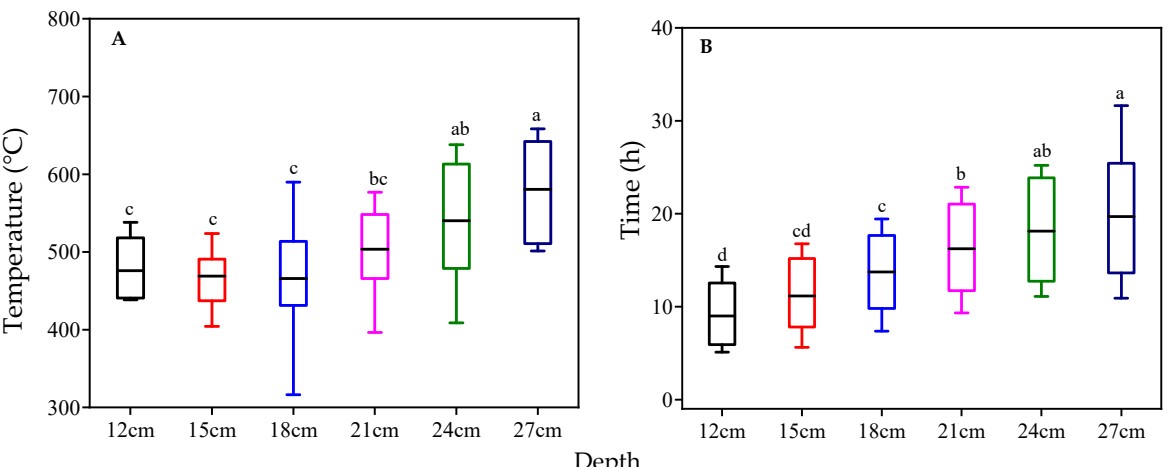

**Figure 7.** Variance analysis on influence of depth on temperature and combustion period of smoldering during sub-surface fires ((**A**): Effect of depth on peak temperature; (**B**): Effect of depth on time; a–d means that if there is any same lowercase letter, the difference is not significant).

The rate of spread of smoldering in humus at 20% moisture content was found to be the fastest, which is significantly greater than the other moisture contents. Although there was no significant difference between the rate of spread in the 30% or 40% moisture content samples, the rate of spread at 40% moisture content was slightly higher than that of 30% (Figure 8).

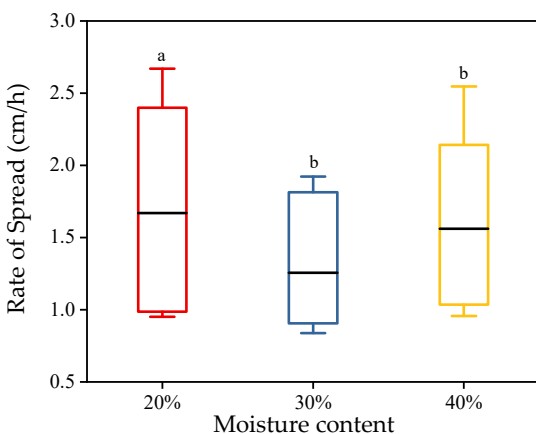

**Figure 8.** Variance analysis on influence of moisture content on rate of spread of smoldering during sub-surface fires (a,b means that if there is any same lowercase letter, the difference is not significant).

For the sample moisture content and burning depth, the regression prediction models of the highest temperatures and rates of spread have been established. Both moisture content and burning depth passed the significance test for the highest temperature using the regression model ($p < 0.01$). The moisture content was negatively correlated with the highest temperature, and the burning depth was positively correlated with temperature. The moisture content passed the significance test of the rate of spread for the regression model ($p < 0.01$) (Table 2).

**Table 2.** Regression fitting of top temperature and spreading rate during the sub-surface fire.

| Dependent Variable | Independent Variable | Coefficient | Standard Error | Significance | *p*-Value | Equation |
|---|---|---|---|---|---|---|
| Peak temperature | Constant | 537.83 | 41.26 | <0.01 | <0.01 | $y_1 = 537.83 - 5.32x_1 + 5.54x_2$ |
| | $x_1$ | −5.32 | 0.94 | <0.01 | | |
| | $x_2$ | 5.54 | 1.32 | <0.01 | | |
| Spread rate | Constant | 4.66 | 0.90 | <0.01 | 0.002 | $y_2 = 4.66 + 0.004x_1{}^2 - 0.22x_1$ |
| | $x_1$ | −0.22 | 0.07 | 0.001 | | |
| | $x_1{}^2$ | 0.004 | 0.001 | 0.002 | | |

$x_1$: moisture content; $x_2$: depth; $y_1$: peak temperature; $y_2$: spread rate.

### 3.3. Occurrence Probability Prediction of Smoldering

Based on the logistic regression model, the occurrence probability prediction model was established for moisture content and burning depth. The results showed that both of these two variables passed the significance test, and the model imitative effect was good ($R^2 = 0.93$), with both moisture content and burning depth being negatively correlated with the occurrence probability (Table 3). The regression equation is as follows:

$$P = \frac{1}{1 + e^{-(75.19 - 1.39x_1 - 1.08x_2)}} \tag{2}$$

where $P$ is the occurrence probability of smoldering; $x_1$ is the moisture content of humus; and $x_2$ is the burning depth.

**Table 3.** Logistic regression fitting of occurrence probability during the sub-surface fire.

| Independent Variable | Coefficient | Standard Error | Significance | $R^2$ |
|---|---|---|---|---|
| Constant | 75.19 | 29.34 | 0.0104 | |
| Moisture content | 1.39 | 0.54 | 0.0095 | 0.93 |
| Depth | 1.08 | 0.47 | 0.0212 | |

The ROC curve was drawn according to the probability predicted by the occurrence probability prediction model. The AUC value of the area under the curve was 0.996, indicating that the prediction accuracy of the model was considered high (Figure 9).

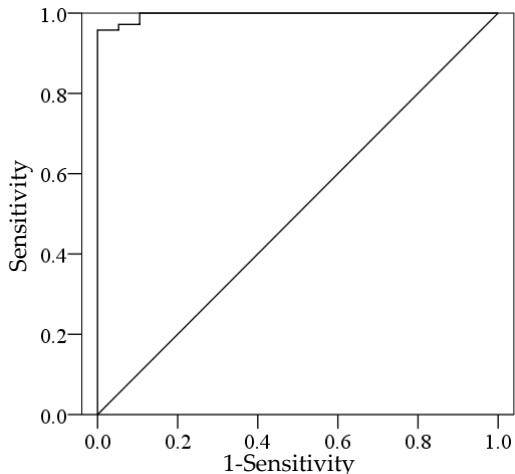

**Figure 9.** ROC curve of the occurrence probability prediction model of smoldering during sub-surface fires.

Table 4 shows the results of the dominance ratio on the occurrence probability model. For humus, moisture content (dominance ratio = 4.008) had a greater impact on the occurrence probability than burning depth (dominance ratio = 2.948). This means that for the same burning depth, the occurrence probability increased 4.008 times when the moisture content decreased by 10%. At the same moisture content, the occurrence probability increased by 2.948 times when the depth decreased by 3 cm.

**Table 4.** Dominance ratio of smoldering occurrence probability by logistic regression (upper and lower limits of 95% confidence interval).

| Independent Variable | Coefficient | Upper Bounds | Lower Bounds |
|---|---|---|---|
| Moisture content | 4.008 | 1.403 | 11.445 |
| Depth | 2.948 | 1.176 | 7.394 |

## 4. Discussion

This study of an indoor small-scale simulated sub-surface fire experiment considered the characteristics of the moisture content limit, depth of burning, and temperature rises in a humus layer of the boreal forests in China. There have been few studies on sub-surface fires of humus in the boreal forest. The characteristics during smoldering are an important indicator, and the moisture content limit is an important condition to the occurrence and spread of sub-surface forest fires [30,36]. In addition, the small-scale simulated smoldering experiment used in this study can greatly simplify our understanding of these complex interactions and can be used to consider the characteristics and basic principles through the control of key variables [37]. Laboratory results should be carefully considered when applying to natural conditions in situ. The results of this study can still provide some insights into the mechanisms of smoldering sub-surface fires in the humus layer of the boreal forest.

### 4.1. Limit Conditions of Smoldering

This study found that when the moisture content of the humus was 40%, the sub-surface fire would be extinguished after a period of time, indicating that 40% was the limit moisture content of sub-surface fires in the humus of the boreal forest. The continuous combustion of sub-surface fires depends on the balance between the energy released

during the combustion process and the energy required for the continued combustion and evaporation of water [38]. Evaporation requires a large amount of energy. The energy released by combustion is not enough to maintain combustion and evaporation at more than 40% moisture content, so it is difficult for combustion to be sustained. Sub-surface fires are easy to ignore in boreal forests because most surface fires are accompanied by sub-surface fires. The sub-surface fires may become extinguished when they spread to deeper layers with higher moisture content, and it is difficult to detect sub-surface fires by the existing forest fire monitoring methods, such as aviation patrol, watchtower monitoring, and artificial ground patrol [39]. This also explains why there are few historical records about sub-surface fires. When the moisture content of humus was 20% and 30%, the sub-surface fires could complete the smoldering process and release a lot of heat. It was generally believed that surface fires were difficult to cause when the moisture content of surface fuel was more than 25%, while sub-surface fires could occur under higher moisture content conditions. Additionally, when the external fire was intense and lasted for a long time, the wet underground combustible material could also be completely dried and smoldering [26]. It also shows that the neglected sub-surface fires could spread for a long period when the moisture content was lower, and the sub-surface fires would not be discovered until a secondary ground fire with significant damage was caused.

Frandsen was the first to determine the limit moisture content conditions of sub-surface fires [40]. The limit moisture content conditions of different types of sub-surface fires were constantly modified with the development of research in recent years [22]. Other studies on peat (duff) found that the moisture content limit of smoldering could reach 50%–60% [12,41], higher than the moisture content limit of humus in this study. The surface fuels under a forest are composed of decomposing and semi-decomposing dead leaves, branches, and barks with large gaps, so the water-retaining ability of these surface fuels is likely to be poor. However, peat has a compact structure with small voids and strong water-retaining ability, so the moisture content is higher than that of surface fuels [38]. At the same time, if materials, whether natural peat or commercial peat, were used with a high content of organic matter [42–44], they are likely to be easier to ignite and spread fire continuously and stably, even at a high moisture content. Garlough et al. [20] also pointed out that the smoldering moisture content limit of the humus layer in the *Pinus ponderosa* forest was 57%. Different ignition methods, sample size and composition, and moisture content calculation methods may all lead to differences in results, which need to be verified by further studies.

*4.2. Characteristics and Factors of Smoldering*

During combustion, the upper layer of humus presents a mixture of ash and carbon black during sub-surface fires, which is the same for surface fires, but smoldering occurs in the lower layers with a slow smoldering rate of spread. When the moisture content was 20%, the fastest rate of spread for peat at a depth of 12 cm was only 1.77 cm/h. This is similar to the findings of Fernandes [45] and Prat-Guitart et al. [41]. At the same time, when sub-surface fires spread to a depth with higher moisture content, time is needed to dry the lower depths, and the combustion temperature is lower during this period [46], which undoubtedly conceals the sub-surface fires, increasing the difficulty of monitoring. When smoldering spreads downward, the upper layer will collapse and lead to increased oxygen availability. As a result, the smoldering in the lower layers will then increase their intensity [47]. The falling burnt material also prevents heat loss, so the temperature of lower layers increases, and the heat loss is reduced. This can keep the temperature of the sub-surface fire above 300 °C for several hours. It is therefore important that when a sub-surface fire occurs, rescue personnel use leg protective clothing, and should take care not to rush into the fireground, even hours after the extinguishment of sub-surface fires.

This study found that the effects of moisture content on the burning depth of humus and their interaction on smoldering temperature and combustion time were significantly different ($p < 0.05$). At the same burning depth, the combustion temperature was highest

when the moisture content was 20%, and the combustion time was long when the moisture content was 30%. The process for sub-surface fires is dehydration and drying, followed by the combustion rate of spread [48,49]. The dehydration of humus with low moisture content requires less energy to ignite and combust, so the combustion temperature is higher; whereas the dehydration of humus with a high moisture content consumes more energy, so the combustion period is longer. For the same moisture content, the combustion temperature and time at different depths vary greatly. A sub-surface fire creeps slowly, and the burning time in the deep layer is naturally the longest. It is worth noting that the temperature changed in a "U" shape with the increase in depth. The temperature was higher at the beginning and end, and lower in the middle at 18 cm. When the moisture content was 40%, the sub-surface fire could only spread to 18 cm and then self-extinguished. This indicates that the depth of 18 cm is critical for sub-surface fires. When humus moisture content cannot be determined, the depth can be used to assist in judging whether a sub-surface fire occurred. Where there is no fire at a depth of 18 cm, this indicates that a sub-surface fire did not occur or has been extinguished. However, sub-surface fires could have spread much deeper. The rate of spread of sub-surface fires was the fastest at 20% moisture content. Although the sub-surface fires could spread to a depth of 18 cm and become extinguished in fuels of 40% moisture content, the rate of spread was slightly higher than that of 30% moisture content, so the moisture content will accelerate the combustion of sub-surface fire [19,50].

*4.3. Occurrence Prediction of Smoldering*

Research on predicting forest fire occurrence has attracted extensive attention in recent years, and the numbers of forest fires dependent on climatic and weather factors, terrain (slope, elevation, and altitude), and fuel types are the basis of typical forest fire occurrence prediction models [51,52]. Due to the concealment of sub-surface fire, it is difficult to obtain data on likely occurrence, which leads to the need for a predictive model. As a result, indirect factors are usually used to predict sub-surface fires, and moisture content, depth, and other conditions are also important factors in the occurrence and development of sub-surface fires [53–55].

Logistic regression models are widely used in the occurrence and prediction of forest fires due to their universality [56,57]. Based on a logistic regression model, this study used two easily obtained variables, moisture content and burning depth, to establish the occurrence probability prediction model of sub-surface fires in the boreal forests of China. The variables both pass the significance testing ($p < 0.05$), and the prediction accuracy of the model is high (AUC = 0.996). Reardon [38,58] also pointed out that the logistic regression model was suitable for the prediction of sub-surface fires, which is confirmed by this study. Humus moisture content had a greater impact on the occurrence probability of sub-surface fire than depth, which indicated that moisture content was the main factor in the occurrence of sub-surface fire. Compared to humus moisture content, it is easier to obtain depth data, due to the variation of moisture content of humus on large spatial scales [55]. The research on the prediction of combustible moisture content has attracted wide attention, and the humus moisture content can be predicted by meteorological elements, topographic data, vegetation coverage, etc. [59–61]. Compared to predicting by experience alone, the occurrence probability prediction model established in this study can provide better support for the monitoring and prediction of sub-surface fires.

## 5. Conclusions

The occurrence of sub-surface forest fires is on the rise, obviously due to the influence of global warming. It is helpful to strengthen and improve the prevention and control technology of sub-surface forest fires by researching the limit moisture content conditions and the occurrence probability. This study quantified that the limit moisture content of sub-surface forest fires is 40% in the humus layer of the boreal forest. When surface fires occur, sub-surface forest fires also easily occur when the moisture content of humus is less

than 40%. Therefore, it is suggested to dig deep fire trenches at the same time to prevent and fight sub-surface forest fires when fighting surface fires. Water is an effective means to inhibit the occurrence and spread of sub-surface forest fires, so it is necessary to spray a large amount of water in the fire trench and humus in order to suppress the fires. The monitoring time of the fire field should be extended after the fire is extinguished due to the high temperature and slow-burning process of the sub-surface fires; probes and iron brazes could also help to monitor sub-surface fires. The fire occurrence probability prediction model based on logistic regression has high accuracy and practical significance for the prediction of sub-surface forest fires.

**Author Contributions:** Conceptualization, Y.S. and S.Y.; methodology, Y.S. and S.Y.; software, Y.S. and S.Y.; validation, Y.S. and S.Y.; formal analysis, Y.S. and S.Y.; investigation, S.Y., B.Y., C.C., and L.C.; resources, Y.S.; data curation, S.Y.; writing—original draft preparation, S.Y; writing—review and editing, Y.S., S.T., and G.D.; visualization, S.Y.; supervision, Y.S.; project administration, Y.S.; funding acquisition, Y.S. All authors have read and agreed to the published version of the manuscript.

**Funding:** This research was supported by The National Natural Science Foundation of China (Grant Nos. 31971669, 32271881).

**Data Availability Statement:** The data presented in this study are available upon request from the corresponding author.

**Acknowledgments:** We thank for the support of Forestry College of Beihua University for the research.

**Conflicts of Interest:** The authors declare no conflict of interest.

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
