# Peer review of "Study on the Limit of Moisture Content of Smoldering Humus during Sub-Surface Fires in the Boreal Forests of China"

_forests, doi:10.3390/f14020252_

Round 1
Reviewer 1 Report
Overall, an interesting paper. I have put specific comments and corrections in the attached .pdf

Reviewer 2 Report
The manuscript entitled Study on the limit of moisture content of smoldering humous during sub-surface fires in the boreal forests of China after Major Revision
In my opinion, the subject of this work is relevant for the Urban Water Journal after Major revisions and approved corrections.
The topic of the paper, is very interesting and important, especially in the context of application and connection of soil moisture and fires in boreal forests.
The Journal Forests MDPI wants interesting and quality papers.
The authors must check word within the title hummus? Is it hummus or humus? Please check.
First, before all the paper has the next sections and sub-sections (i.e. Abstract, Introduction, Methods, Study area, Sampling and processing, Simulating smoldering experiment, Data processing and analysis, Results, Limit moisture content of sub-surface fire smoldering, Factors of smoldering, Occurrence probability prediction of smoldering, Conclusion and discussion, Limit conditions of smoldering, Characteristics and factors of smoldering, Occurrence prediction of smoldering, etc.)
I strongly recommend to the authors add a section of abbreviation because of the large number of short words (terms).
The section of Abstract
The section of abstract is not fully explaining the main results of this research. The authors must add more sentences that reflect moisture content in connection with fires in boreal areas.
Also the authors must add few sentences which explain the main results of this research. In this section it is necessary to explain main goal and expectations of this research.
The section of Introduction
This section is overall short and must be extended with more references. How is the sub-surface layer of humus and water content measured? There are plenty of methodologies that have can explain better soil moisture or water content in the land. The authors in the section Introduction must explain better how is possible to measure soil moisture. We have for example good Remote Sensing methods and procedures for measuring of soil moisture.
I strongly recommend two references to be cited.
- Valjarević, A, Filipović, D, Valjarević, D, et al. GIS and remote sensing techniques for the estimation of dew volume in the Republic of Serbia. Meteorol Appl. 2020; 27:e1930. https://doi.org/10.1002/met.1930.
- MacDonald, L.H. and Huffman, E.L. (2004), Post-fire Soil Water Repellency. Soil Sci. Soc. Am. J., 68: 1729-1734. https://doi.org/10.2136/sssaj2004.1729.
In this section, the authors can add more about the connection between hummus, soil and moisture but in regional and global scale. This part also can compare another research previously published.
Section Methods
The authors tried to explain the research area. I recommend to valuable method to do this.
The first is to draw a map or to define geographical coordinates within the text.
The second is to define the geographical coordinates of the research area.
In this section it is necessary to add more sentences about regional climate patterns of the investigated area.
Section Sampling and processing
Line 94 and 95, why the author used these dimensions of data (30m×20m and 50cm×50cm) ?
Line 100 check the reference [2020]. I think it is a fault.
Figure 1, This Figure must be explaining better procedure?
Equation 1, why the authors used Natural Logarithm it is not clearly for me?
In the beginning of the text the authors explained that used significant value of test (P<0.01).
Now the authors labeled the another value (P<0.05), why?
The similar test of statistical significance is ANOM, can the authors compare this test with ANOVA. For this calculations it is necessary to compare various methodologies.
Figure 3. Can the authors explain how they estimated moisture content and is it this Figure in scale or not?
Figure 5, how many samples used for this result?
Equation 2, what is the burning depth? Please explain better.
The Logistic regression model is not usual used for analyzing ow soil properties? Please explain better, why the authors used this model?
The main problem within this manuscript is missing the section of Discussion. This section must be mandatorily written. In this section, the authors must add more sentences (literature) of previously published research.
The section Conclusion is overall short.
In this section, the authors must answer on following questions?
Why this research is important?
Did the authors find a connection between soil moisture content and sub-surface fires in boreal forests?
Please, add more results and main goals in the section of Conclusion.
This paper has the potential to be published. The authors did a lot of things within this manuscript. The paper is very interesting and scientifically correct.
In the end, I recommend Major Revision.
Good luck to the authors
The Reviewer#2

Reviewer 3 Report
The author's topic is very interesting, relevant to the efforts to suppress forest fires, and in accordance with the scope FOREST. However, this manuscript must be corrected to be suitable for subsequent processes, especially in the introduction and method sections. Citations should be added, especially in the method section.
Section 1. Introduction: The objectives of this research have not been clearly stated.
Section 2.1: the author should describe the fire incidents in Bila River National Nature Reserve, how the condition of the humus thickness, the density, etc.
Section 2.2: How many 30m×20m plots are used? How is it placed in the field? Add an image showing the placement of quadrats of 50cm×50cm in a plot.
Line 127-128: why did the authors not consider the sample density placed into the smoldering furnace?
Section 2.4: Please add citations for the method used, including the formula (1).
Line 267: Conclusion should be separated from the discussion. Based on the results, what recommendations can be suggested to stakeholders to suppress the occurrence of sub-surface fires?
Round 2
Reviewer 2 Report
The manuscript entitled Study on the limit of moisture content of smoldering humous during sub-surface fires in the boreal forests of China can be accepted in its present form. The authors answered all of my comments and corrected all of the mistakes within the text.
In my opinion, this manuscript can be accepted in its present form.
Sincerely,
The Reviewer #3

Author Response
Thank you very much for your valuable suggestions, which have greatly helped me improve my paper.
Thank you!Good luck!
Reviewer 3 Report
The authors have made improvements as suggested, but a few things still need to be added:
Point 3: Section 2.2: How many 30m×20m plots are used? How is it placed in the field? Add an image showing the placement of quadrats of 50cm×50cm in a plot.
Response 3: A total of 5 30m×20m sample plots have been set, and pictures have been added to the text to indicate the position of 50cm×50cm.
Reviewer: Have you included the plot number in the manuscript?
Point 5: Section 2.4: Please add citations for the method used, including the formula (1).
Response 5: A reference to the usage method has been added to the article.
Reviewer: Line 157-166, 169-174: need citation
The list of acronyms should not be in the form of a table but can be placed as an appendix.
Author Response
Point 1: Have you included the plot number in the manuscript?
Response 1: The number of sample plots has been added in the Sampling and processing section.
Point 2: Line 157-166, 169-174: need citation.
Response 2: References have been added to the paper.
Point 3: The list of acronyms should not be in the form of a table but can be placed as an appendix.
Response 3: Acronyms table has been deleted in the paper.